# Analysis of Genomic Alternative Splicing Patterns in Rat under Heat Stress Based on RNA-Seq Data

**DOI:** 10.3390/genes13020358

**Published:** 2022-02-16

**Authors:** Shangzhen Huang, Jinhuan Dou, Zhongshu Li, Lirong Hu, Ying Yu, Yachun Wang

**Affiliations:** 1National Engineering Laboratory of Animal Breeding, Key Laboratory of Animal Genetics, Breeding and Reproduction, MARA, College of Animal Science and Technology, China Agricultural University, Beijing 100193, China; hsz19980225@163.com (S.H.); b20193040324@cau.edu.cn (L.H.); yuying@cau.edu.cn (Y.Y.); 2Animal Science and Technology College, Beijing University of Agriculture, Beijing 100193, China; 3Agricultural College, Yanbian University, Yanji 133002, China; lizhongshu@ybu.edu.cn

**Keywords:** heat stress response, post-transcriptome, alternative splicing, liver, rat

## Abstract

Heat stress is one of the most severe challenges faced in livestock production in summer. Alternative splicing as an important post-transcriptional regulation is rarely studied in heat-stressed animals. Here, we performed and analyzed RNA-sequencing assays on the liver of Sprague-Dawley rats in control (22 °C, *n* = 5) and heat stress (4 °C for 120 min, H120; *n* = 5) groups, resulting in the identification of 636 differentially expressed genes. Identification analysis of the alternative splicing events revealed that heat stress-induced alternative splicing events increased by 20.18%. Compared with other types of alternative splicing events, the alternative start increased the most (43.40%) after heat stress. Twenty-eight genes were differentially alternatively spliced (DAS) between the control and H120 groups, among which *Acly*, *Hnrnpd* and *mir3064* were also differentially expressed. For DAS genes, *Srebf1*, *Shc1*, *Srsf5* and *Ensa* were associated with insulin, while *Cast*, *Srebf1*, *Tmem33*, *Tor1aip2*, *Slc39a7* and *Sqstm1* were enriched in the composition of the endoplasmic reticulum. In summary, our study conducts a comprehensive profile of alternative splicing in heat-stressed rats, indicating that alternative splicing is one of the molecular mechanisms of heat stress response in mammals and providing reference data for research on heat tolerance in mammalian livestock.

## 1. Introduction

The non-specific response of an organism caused by excessively high environmental temperature is called heat stress [1] and it usually occurs when the ambient temperature is above the upper critical temperature of the thermal neutral zone. The efficiency of livestock products is compromised under heat stress conditions, since nutrients are diverted to maintain euthermia [2]. The cumulative impacts of heat stress on feed intake, metabolism and physiology status may result in reduced milk yield in dairy cattle [3], decreased body weight and growth rate in broiler [4], pig [5], lamb [6] and beef cattle [3]. Furthermore, heat stress has detrimental effects on both male and female reproductive functions [7] and threatens animal health and welfare [8]. According to a survey, heat stress has placed a huge economic burden on the animal husbandry industry [9], e.g., in the United States alone, the loss of dairy induced by heat stress is approximately $900 million/year and the loss of beef and swine exceeds $300 million/year.

To the best of our knowledge, the temperature–humidity index (THI) is usually used as the environmental index in heat stress studies [10]. Different species have different THI thresholds for determining the occurrence of heat stress [11]. The THI reflects the heat stress of a population from an environmental perspective, but at an individual level, few accurate markers have been used in the prediction of heat stress, which may be largely related to the fact that the molecular mechanism regulating heat stress is not clear. With the development of next-generation sequencing technology, numerous studies (including our previous studies) have identified lots of heat stress-related genes using RNA-sequencing (RNA-Seq) [12,13,14,15,16,17], but transcriptome information is still incomplete. A comprehensive analysis of alternative splicing of transcripts is lacking. The phenomenon of alternative splicing of genes was first proposed in 1978, that is, a pre-mRNA generates multiple different mRNA isoforms by selecting different splicing sites [18]. Alternative splicing is a ubiquitous mechanism in higher eukaryotes and contributes to both transcriptome and proteome diversity [19,20]. In addition, alternative splicing is involved in many physiological processes, as well as responses to biotic and abiotic stresses [21,22,23,24,25].

Studies on alternative splicing of heat-stressed animals have been carried out. As early as 1994, Takechi et al. detected that heat stress caused alternative 5′ splice site selection of *HSP47* in mice [26]. The activation of potential alternative 5′ splice sites induced by heat stress is widespread in the human genome [27] and regulated by the suppression of splicing mechanisms [28]. The ratio of alternative splicing isoforms of *TLR4* in Bama minipig (*Sus scrofa domestica*) [29] and *dHSF* in Drosophila [30] are changed after heat stress. Kaitsuka et al. found that eEF1Bδ changes its structure and function from a translation factor into a heat-shock response transcription factor by alternative splicing and induces heat-shock element (HSE)-containing genes [31]. In addition, Tan et al. conducted the first comprehensive study on alternative splicing in heat-stressed fish in 2019 [32]. As far as we know, there has not been a comprehensive transcriptome study to identify heat stress-induced alternative splicing changes in mammals.

Based on the heat-stressed rat model that was built in our previous study [14], a comprehensive analysis of alternative splicing rules was performed in the current study. After the bioinformatic analysis, the differentially expressed genes (DEGs) and differentially alternatively spliced (DAS) genes were identified and important biological processes involved in the heat stress response were analyzed. This work further provides reference data for research on heat tolerance in mammalian livestock.

## 2. Materials and Methods

### 2.1. Animals and Treatments

The in vivo rat experiments were performed at the College of Animal Science and Technology, China Agricultural University. The Institutional Animal Care and Use Committee approved all the experimental procedures, which complied with the China Physiological Society’s guiding principles for research involving animals and adhered to the high standard (best practice) of veterinary care as stipulated in the Guide for Care and Use of Laboratory Animals.

In previous research [14], 99 eight-week old female specific-pathogen-free (SPF) Sprague-Dawley rats (Beijing Vital River Laboratory Animal Technology Co., Ltd., Beijing, China) weighing 205 ± 7.16 g were used as subjects. Prior to the experiments, a total of three rats per cage were housed in Nalgene polycarbonate cages (40 × 30 × 180 cm^3^, Beijing Vital River Laboratory, Animal Technology Co, Ltd., Beijing, China) at 22 ± 1 °C and 50% relative humidity (RH) with a 12 h reverse light/dark cycle (on 06:00, off 18:00) for one week. Rats were given access to feed and water ad libitum and all experiments were conducted with healthy and conscious rats. As previously described [14], five rats randomly assigned to the heat stress group were exposed to 42 °C for 120 min (H120) and five rats in the control group were reared at 22 ± 1 °C. The heating experiments were completed in a floor-standing artificial climate incubator (BIO250, BOXUN Medicine Instrument Co, Shanghai, China). Rats in the control group were never introduced into the incubator. After treatments, the rats were anesthetized by intraperitoneal injection of 1%, 1.2 mL sodium pentobarbital (40 mg/kg of body weight) and liver tissues were collected. Liver tissues were washed in ice-cold phosphate buffer solution (PBS) and snap-frozen immediately in liquid nitrogen until further analysis.

### 2.2. RNA Extraction, cDNA Library Construction and Illumina Deep Sequencing

Total RNA was extracted from quick-frozen liver samples using TRIzol^®^ reagent (Invitrogen Life Technologies, Palo Alto, CA, USA), according to the manufacturer instructions. After homogenising the sample with TRIzol^®^ Reagent, chloroform was added, RNA was precipitated with isopropanol, then treated with 75% ethanol [33,34]. The total RNA was dissolved using DNase/RNase-free water. The RNA quality and quantity were determined using agarose gel electrophoresis and NanoDrop 2000 (Thermo Fisher Scientific, Waltham, MA, USA). The RNA integrity was assessed using the RNA Nano 6000 Assay Kit in the Agilent Bioanalyzer 2100 system (Agilent Technologies, Santa Clara, CA, USA). Then, the mRNA was purified and enriched from total RNA using Poly-T oligo-attached magnetic beads and sheared into fragments. First-strand cDNA was generated using random hexamer primers and M-MuLV Reverse Transcriptase (RNase H). Second-strand cDNA was synthesized using DNA polymerase I and RNase H. The library fragments were purified with the AMPure XP system (Beckman Coulter, Beverly, MA, USA) to select cDNA fragments approximately 200 bp in length and PCR amplified. Finally, the library was sequenced in paired-end 150 bp reads using the Illumina^®^ HiSeq 2000 platform. 

### 2.3. Data Filtering and Transcriptome Alignment

The raw reads were filtered by removing the adapter sequences; reads with poly-N and low-quality reads (i.e., more than 50% of the reads with a quality score under 10 or read length <30) were trimmed using Trim Galore 0.4.5 (http://www.bioinformatics.babraham.ac.uk/projects/trim_galore/ (accessed on 12 February 2020)) and FastQC 0.11.8 [35] software. In addition, Q20, Q30 and GC content were calculated to evaluate data quality. High-quality reads were aligned to the rat reference genome (*Rattus norvegicus* 6.0.98, Rnor 6.0.98) using STAR 2.5.3 [36] with the default parameters. Only reads uniquely aligned to the reference genome were used for downstream analysis.

### 2.4. Identification of Differential Expression Genes

The aligned reads were assembled using the StringTie 1.3.5 [37]. The stattest function in Ballgown 3.11 [38] was applied to identify DEGs between the Control and H120 groups. The expression levels of the genes were normalized with FPKM (fragments per kilobase per million mapped fragments). Genes with BH-adjusted *p*-value (*q*-value) < 0.05 and |log_2_FoldChange| > 1 were considered as DEGs.

### 2.5. Identification of Differential Alternative Splicing Events

Alternative splicing patterns were analyzed using SGSeq 1.22.0 [39] via estimating the percent spliced in (PSI) for each variant, taking “TxDb.Rnorvegicus.UCSC.rn6.refGene” as the reference genome. A PSI greater than 0 and less than 1 indicates that this alternative splicing event occurred on the sample level. The *t*-test was used to compare the difference in the number of events between the Control and H120 groups. For each group, the alternative splicing events that occurred in at least three samples were retained at the group level. Eight types of alternative splicing events were distinguished, including alternative 5′ splice site (A5SS), alternative 3′ splice site (A3SS), skipped exon (SE), retained intron (RI), mutually exclusive exons (MXE), alternative start (AS), alternative end (AE) and unknown type.

The DEXSeq 3.11 [40] was used to perform a statistical test for differential variant usage between the Control and H120 groups. A BH-adjusted *p*-value (*q*-value) < 0.05 and differences in PSI values (dPSI) between conditions > 0.1 were set as criteria to filter DAS events. Genes with at least one DAS event were determined as DAS genes.

### 2.6. Integrated Gene Expression and Alternative Splicing Results

The clustering analysis of DEGs and DAS genes was performed using Venn2.1 (https://bioinfogp.cnb.csic.es/tools/venny/ (accessed on 28 August 2021)). The overlapped genes were considered as DAS genes with different expression levels between the Control and H120 groups. A hypergeometric distribution test was performed in order to test whether the overlap was significant (https://systems.crump.ucla.edu/hypergeometric/ (accessed on 28 August 2021)). In order to obtain protein structure information (tertiary structure score) and functional information (number of functional residues and domain score) of the overlapping genes, the APPRIS Database [41] was employed to annotate the alternative splicing isoforms of these genes.

### 2.7. Functional Analysis of Differentially Expressed Genes and Differentially Alternatively Spliced Genes

The functional enrichment analysis of DEGs and DAS genes considering Gene Ontology (GO) terms and the Kyoto Encyclopedia of Genes and Genomes (KEGG) pathway was conducted using DAVID v.6.8 [42]. Results with a false discovery rate (FDR) value < 0.05 for DEGs and a *p*-value < 0.05 for DAS genes were considered as significant. The functions of DAS genes with significant different expression levels were further searched in the UniProt Knowledgebase [43] and the Rat Genome Database [44].

### 2.8. Validation of the Expression Level of the DAS Genes by Real-Time Quantitative PCR

The RNA of the liver was transcribed into cDNA using the PrimeScript RT reagent Kit with gDNA Eraser (Takara). The primers of five randomly selected DAS genes are listed in Appendix A, and the glyceraldehyde-3-phosphate dehydrogenase gene (*GAPDH*) was used as the internal reference gene. Primers for *GAPDH* and DAS genes were designed using Primer-BLAST (https://www.ncbi.nlm.nih.gov/tools/primer-blast/ (accessed on 30 December 2020)) [45], and primers were synthesized by Beijing Tsingke Biological Technology (Beijing, China). Each reaction was performed in 20 mL mixtures, including a 2 mL diluted cDNA sample as template, 10 mL SYBR Green I Master (Roche), 0.6 mL forward and 0.6 mL reverse gene-specific primers, and 6.8 mL ddH2O. Amplification conditions were set as follows: denaturation at 95 °C for 10 min, followed by 40 cycles of 98 °C for 10 s and 60 °C for 20 s for annealing, and extension at 72 °C for 20 s, followed by a final extension at 65 °C for 1 min. Triplicate real-time quantitative PCRs (RT-qPCR) were accomplished for each cDNA sample. The comparative threshold cycle (Ct) value method was adopted to analyze relative gene expression. The Pearson correlation coefficients between the FPKM counts from the RNA-Seq analysis and expression levels relative to *GAPDH* from the RT-qPCR analysis were calculated.

## 3. Results

### 3.1. Summary of the Basic Information

The average of the total paired-end raw reads of the samples was 31,106,148 bp (ranging from 26,255,321 to 38,816,107 bp), and the average of high-quality clean data after data filtering was 30,565,891 bp with a GC content of 49.73% (Appendix A). The average percentages of Q20 and Q30 were 94.95% and 89.34%, respectively. After alignment analysis, the total mapped rates ranged from 94.69% to 97.88% among all samples. Bioinformatic analysis for high-throughput transcriptome data revealed a total of 10,707 genes annotated in the liver tissues.

### 3.2. Identification of Differentially Expressed Genes

A total of 636 DEGs were identified in the comparison of Control vs. H120, among which 374 genes were upregulated and 262 genes were downregulated (Figure 1 and Appendix A). The top five DEGs includes two upregulated genes (*Mical2* and *Arl5b*) and three downregulated genes (*Paqr7*, *Rmnd5b* and *Alpl*). There were seven DEGs with |log_2_FoldChange| > 3.5, among which the expression levels of *Hspa1b* and *Hspb1* in the H120 group were 152.63 and 11.53 times higher than those in the Control group, respectively.

### 3.3. Functional Enrichment Analysis for Differentially Expressed Genes

Functional enrichment analysis of DEGs revealed that a total of eleven biological processes (BP), six cellular components (CC) and one molecular function (MF) were significantly enriched (FDR value < 0.05) (Figure 2 and Appendix A). Genes such as *Dnaja1*, *Socs3*, *Hspa8*, *Hsp90aa1*, *Eif2b3*, *Abcc2*, *Pklr*, *Dnaja4*, *Tp53inp1*, *Loc103692716* and *Hspa1b* were found to be related to the biological process of response to heat (GO:0009408). Furthermore, thirty-nine genes had the function of regulating the oxidation–reduction process (GO:0055114). In total, ten KEGG pathways were significantly enriched (FDR value < 0.05) (Figure 2 and Appendix A), including biosynthesis of antibiotics (rno01130) as the most significant pathway. A total of 62 DEGs were involved in metabolic pathways (rno01100), especially in pyruvate metabolism (rno00620), carbon metabolism (rno01200) and fatty acid metabolism (rno01212).

### 3.4. Summary of Alternative Splicing Events

The number of alternative splicing events that occurred in each sample is shown in Figure 3. At the sample level, an average of 273 and 311 alternative splicing events were identified from the Control and H120 groups, respectively. The difference between the Control and H120 groups was not significant (*p* = 0.1). At the group level, a total of 228 alternative splicing events occurred in the liver of the Control, corresponding to 220 genes; 274 alternative splicing events occurred in the liver of the H120, corresponding to 256 genes, showing that heat stress-induced alternative splicing events increased by 20.18%, the corresponding alternatively spliced genes increasing by 16.36%.

Eight types of alternative splicing events were identified in the Control and H120 groups (Table 1). In detail, AS accounts for the highest proportion of the two groups (23% in Control and 27% in H120), followed by SE (20% in Control and 21% in H120). The proportion of other types of events is between 10% and 15%, except for MXE and an unknown event at less than 5%. The number of all types of alternative splicing events increased after heat stress except for RI. AS increased the most after heat stress, from 53 to 76.

### 3.5. Identification of Differential Alternative Splicing Events and Genes

Forty alternative splicing events had a *q*-value < 0.05, including 26 events with dPSI > 0.1, which were considered to be DAS events (Table 2), with six AS, five AE, four SE, four RI, three A5SS, two A3SS, as well as one each for MXE and unknown. A single variant of some DAS genes (*Cast*, *Hnrnpf*, *Srsf5*, *Hnrnpd*, *Crem* and *Zmynd11*) corresponds to multiple transcripts, indicating that there was at least one alternative splicing event in other positions of the gene. The *Ngrn*’s alternative splicing event produces non-coding transcripts. It can be seen from the Figure 4a that the samples of each group were clustered together according to PSI, and there were significant differences between the Control and H120 groups. The dPSI values of *Abcg3l2*, *Abcg3l4*, *Tor1aip2*, *Cast* and *Zmynd11* were all greater than 0.35. The |log_2_FoldChange| of five DAS genes in the H120 and Control groups generated by RT-qPCR were in line with the results of the RNA-Seq data (Figure 4b). The Pearson correlation coefficient between RT-qPCR and RNA-Seq was as high as 0.95, which confirmed the reliability of the RNA-Seq analysis.

### 3.6. Functional Enrichment Analysis for Differentially Alternatively Spliced Genes

A total of 14 BPs, 6 CCs, 7 MFs and 0 pathways were annotated, of which 7 BPs, 5 CCs and 4 MFs were significantly enriched (*p*-value < 0.05), as shown in Table 3. Liver development (GO:0001889) containing five genes was the most significantly enriched (*p*-value < 0.001). Three significantly enriched BPs (GO:0032868, GO:0032869 and GO:0050796) were all related to insulin, containing four DAS genes (*Srebf1*, *Shc1*, *Srsf5* and *Ensa*). Negative regulation of transcription from the RNA polymerase II promoter (GO:0000122) was the only BP that was significantly enriched in both DEGs and DAS genes.

### 3.7. The Differentially Alternatively Spliced Genes with Different Expression Levels

Three DAS genes were also differentially expressed between the Control and H120 groups, accounting for 10.71% of DAS genes and 0.47% of DEGs. The hypergeometric distribution test showed that the overlap of DEGs and DAS genes was not significant (*p* = 0.23). The expression levels of *Acly* and *Hnrnpd* after heat stress (Figure 5a) were downregulated to 0.23 and 0.41 times that of the Control, respectively. The expression level of *mir3064* was upregulated to 2.35 times that of the Control. The average per-base read coverages and splice junction counts demonstrated that, after heat stress, the skipping ratio of the fourteenth exon of *Acly* and the skipping ratio of the second exon of *Hnrnpd* increased by 0.33 and 0.23 times (Figure 5b,c), respectively, and the gene expression level decreased significantly (*q*-value < 0.05).

Annotated alternative splice isoforms in the APPRIS Database, the two transcripts of *Acly*, including NM_016987 and NM_001111095, encode proteins with different amino acid lengths (1101 vs. 1091) and tertiary structure scores (2109.4 vs. 2067.2). The amino acid lengths of the four protein isoforms of *Hnrnpd* from long to short are 353, 334, 304 and 285, corresponding to the four transcripts NM_024404, NM_001082539, NM_001082540 and NM_001082541, respectively, and their tertiary structure scores were 413.2, 425.3, 405.08 and 417.27, respectively. The number of functional residues and domains scores of the protein isoforms of *Acly* and *Hnrnpd* have not changed, which means that alternative splicing has no effect on the function of the encoded protein.

The ATP citrate synthase encoded by *Acly* catalyzes the cleavage of citrate into oxaloacetate and acetyl-CoA, the latter serving as a common substrate for de novo cholesterol and fatty acid synthesis. *Acly* is involved in the biosynthetic processes of lipids and fatty acids and also participates in the metabolic processes of acetyl-CoA, citrate and oxaloacetate. Heterogeneous nuclear ribonucleoprotein D0 (hnRNP D0) encoded by *Hnrnpd* binds with high affinity to RNA molecules that contain AU-rich elements found within the 3′-UTR of many proto-oncogenes and cytokine mRNAs. hnRNP D0 can also bind to double- and single-stranded DNA sequences in a specific manner and functions as a transcription factor. The *Hnrnpd* can regulate gene expression at the level of transcription and translation. In addition, *Hnrnpd* plays an important role in liver development and the regulation of circadian rhythms, besides being related to cellular responses to estradiol stimuli and organonitrogen compounds.

## 4. Discussion

Heat-stress experiments in rats allow for easier control of experimental conditions and shorter experimental times. Based on a preliminary exploration [46], this study selected conditions of 42 °C, 50% RH (THI = 93.96) for 120 min with the strongest heat stress performance in order to simulate the physiological response state of rats under short-term, non-extreme lethal heat stress conditions [47,48,49]. In livestock production, the daily RH of the farm is constantly changing, and THI can more easily quantify the degree of heat stress experienced by herds. Studies have made efforts to determine the THI threshold of heat stress and the threshold has been reported variably given different physiological parameters and different production systems [50]. In the study of model animals, specific temperatures and RHs can better reflect the exact environment and contribute to the exploration of heat stress response mechanisms at the level of control variables. The flexible time setting also provides more possibilities to reveal the process of heat stress response. Studies have compared blood indicators, production performance and gene expression levels during acute and chronic heat stress [49,51,52]. Although different studies have different divisions of time, they all provide a reference for understanding the response state that changes over time during heat stress. 

In this study, the liver was selected for a global transcriptome analysis to study alternative splicing induced by heat stress. The liver plays a major role in the metabolic regulation and energy balance of the stress response [53]. Therefore, studying the effects of heat stress on the liver transcriptome can help reveal the effects of heat stress on body metabolism and other aspects. To date, studies have found that heat stress has various effects on the liver. Hundreds of liver genes were differentially expressed after heat stress in different animals, including rabbits [16], fish [53,54], broilers [55] and mice [56], and the expression levels changed differently over time [57]. In addition, the alternative splicing of liver genes in fish was also affected by heat stress [32,58]. Hepatic proteins involved in the processes of heat shock response, immune defense and oxidative stress response were found to be differentially abundant when comparing heat stress with the thermal neutral zone [59]. It is reported that heat-stressed cattle have reduced albumin secretion and liver enzyme activities [60]. Previous studies have also indicated that the liver has a higher maximum temperature than other organs during heat stress [61,62]. Liver damage caused by heatstroke is manifested as centrilobular degeneration or necrosis of hepatocytes and congestion [63]. Therefore, it is certain to have been worthwhile to select the liver for further study here.

The number of alternative splicing events and alternatively spliced genes increased after heat stress in this study, which was consistent with the research performed with fish [32]. These stress-induced increases in alternative splicing were also observed in plants [64,65] and shrimp [25]. Studies have shown that alternative splicing can improve plant tolerance to environmental stress by increasing proteome diversity [66,67]. The increase of alternative splicing may be caused by splicing errors [68,69]. Most abnormal splicing events (splicing errors) can be eliminated by mRNA monitoring mechanisms, such as nonsense mediated decay (NMD) [70,71]. Among DAS genes identified in this study, *Ngrn* produced a non-coding mRNA (NR_028055.1) whose proportion increased due to heat stress and is also a candidate for NMD [72]. *Ngrn* regulates mitochondrial 16S rRNA and intra-mitochondrial translation and is essential for oxidative phosphorylation [73]. Liver tissue may be able to increase the abundance of proteins by increasing alternative splicing to cope with heat stress, together with the mRNA monitoring mechanisms.

In this study, a total of 28 DAS genes associated with heat stress were identified and 10.71% of genes (*Acly*, *Hnrnpd* and *mir3064*) also showed significant differences in transcriptional expression (*q*-value < 0.05 and |log_2_FoldChange| > 1). This ratio was 13.4% in response to heat stress in tea leaves [24]. Therefore, for most genes, changes in alternative splicing are not the main cause of changes in gene expression and changes in alternative splicing do not necessarily lead to changes in gene expression. In *Arabidopsis thaliana*’s response to cold stress [67] and cotton’s response to salt stress [64], it was also found that about two-thirds and four-fifths of differential alternative splicing did not cause differential gene expression, respectively. Zhu et al. [64] believed that the alternative splicing had an independent regulation pattern different from transcriptional regulation. These results indicate that differential expression analysis of genes and various regulatory mechanisms should be integrated to reveal the complex regulation mechanism of the heat stress response.

Among the CCs enriched by DAS genes, the endoplasmic reticulum (ER), containing six genes (*Cast*, *Srebf1*, *Tmem33*, *Tor1aip2*, *Slc39a7* and *Sqstm1*), was most significant. ER homeostasis can be perturbed by heat stress, resulting in ER stress [74,75], and protein processing in the ER pathway is critical for the heat-stress response [54]. This study identified four DAS genes (*Srebf1*, *Shc1*, *Srsf5* and *Ensa*) whose main functions are related to the response to insulin and regulation of insulin secretion. The increase in plasma insulin concentration after heat stress has been confirmed in cows [76], pigs [77] and rodents [78]. Heat stress-specific insulin increase appears to be adaptive and protective in nature towards stressors [2,79]. These genes may be involved in the heat stress response through alternative splicing changes.

ATP citrate lyase, encoded by *Acly*, is an important enzyme in controlling substrate supply for lipid synthesis de novo [80] and is upregulated to different degrees in many kinds of cancers [81]. The depletion of ATP citrate lyase suppressed tumor growth [82], so it has been identified as a potential molecular target for cancer therapy [83]. In this study, heat stress downregulated the expression level of DAS gene *Acly.* Similar results were found in chicken liver at the gene expression level, but heat stress increased the protein level of *Acly* [84]. However, in the heat stress of rabbits [16] and mice [56], the expression level of liver *Acly* is upregulated. Yadav et al. [85] found that the expression levels of *Acly* were regulated in different directions in different germ cells and identified *Acly* as a potential heat-sensitive target in germ cells. The *Acly* protein level in the rumen tissue of lactating dairy cows increased after chronic and mild heat stress [86]. In addition, Moon et al. [87] found that the ratios of the two mRNA isoforms of *Acly* were the same among tissues in rats and proposed that *Acly*’s alternative splicing may be related to various metabolic diseases. This study proves that *Acly*’s alternative splicing was also significantly changed by heat stress, at the gene-expression level as well.

*Hnrnpd* was downregulated after heat stress in this study, as was found in limpet foot tissue [17]. The hnRNP D0 encoded by *Hnrnpd* serves as a key factor involved in mRNA decay, and there exists a significant difference among the stabilizing effects of the four isoforms [88]. hnRNP D0 may also be implicated in endothelial cell senescence [89]. Moreover, hnRNP D0 exhibits a strong response to oxidative stress; the protein decreased rapidly when cells were exposed to hydrogen peroxide [90]. As a main splicing factor involved in the formation and regulation of alternative splicing [91], *Hnrnpd* may in turn regulate the splicing of other pre-mRNAs and cause the difference of gene expression between the H120 and Control groups. *Hnrnpd* may be the predominant link between splicing regulation and heat stress response in rat livers. miRNAs play an important role in the transcriptional regulation of genes coding for proteins involved in heat-stress response-related mechanisms [92,93]. In this study, the expression of *mir3064* was significantly increased under high temperature, consistent with results for laying hens [94]. miR-3064-5p has been reported to inhibit cell proliferation and invasion in ovarian cancer [95] and to suppress angiogenesis in hepatocellular carcinoma [96]. Furthermore, a tight connection between induced oxidative stress and changed *mir3064* expression was observed in retinal pigment epithelial cells under oxidative stress conditions, suggesting that miR-3064 is a stress-responsive factor [97]. The target genes of *mir3064* regulating heat stress deserve further study.

In this study, we did not consider the unfixed duration of heat stress, the fluctuating temperature and the effect of cooling facilities, but our research results may provide a reference for research in more complex situations. Further study needs to be performed to investigate the alternative splicing events of rats exposed to different heat-stress durations and intensities. Although the liver is one of the main organs involved in the response to heat stress, it is still necessary to analyze alternative splicing event in other tissues, such as blood and adrenal glands, under heat stress to explore the mechanism of the stress response more comprehensively.

## 5. Conclusions

This study analyzed the changing pattern of alternative splicing in rat liver tissue under heat stress. The number of alternative splicing events under heat stress increased by 20.18%. Twenty-eight DAS genes were identified, and the molecular functions are mainly enriched in liver development and response to insulin. Three DAS genes (*Acly*, *Hnrnpd* and *mir3064*) which were differentially expressed between the Control and H120 groups can be considered as candidate markers for heat stress in rats. Taken together, these findings indicate that alternative splicing is one of the molecular mechanisms of heat stress responses in mammals.

## Figures and Tables

**Figure 1 genes-13-00358-f001:**
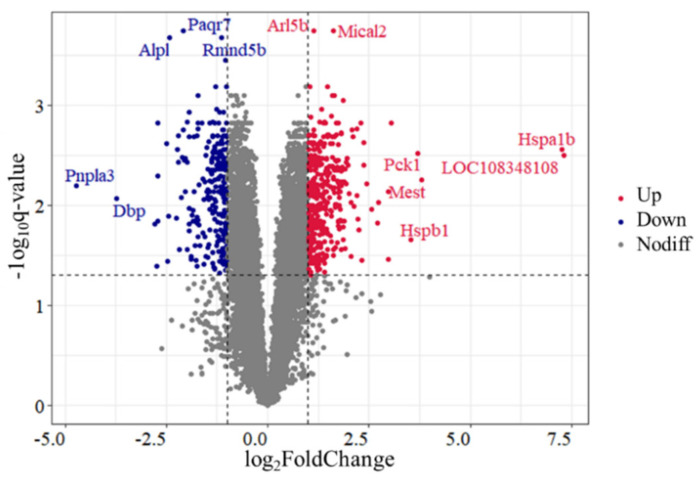
Volcano plot of differentially expressed genes (DEGs) identified in rat liver tissues in the Control and H120 groups. Up and Down represent that the expression levels of DEGs were significantly (*q*-value < 0.05 and |log_2_FoldChange| > 1) higher and lower in the H120 group compared with the Control, respectively. Nodiff means the expression levels of genes are not significantly different between the Control and H120 groups. The DEGs with a *q*-value < 0.0003 or |log_2_FoldChange| > 3.5 were marked with gene names.

**Figure 2 genes-13-00358-f002:**
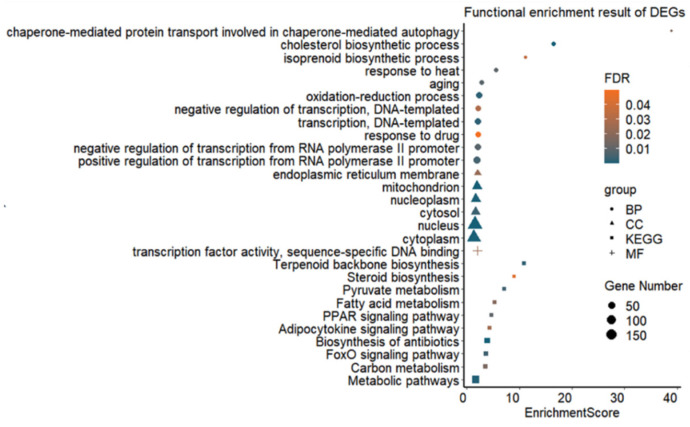
Significantly enriched GO terms and pathways for DEGs in rat liver tissues in the Control (22 °C, *n* = 5) and heat stress (42 °C for 120 min, H120; *n* = 5) groups. BP = biological process, CC = cellular component, MF = molecular function, KEGG = Kyoto Encyclopedia of Genes and Genomes pathway, FDR = false discovery rate.

**Figure 3 genes-13-00358-f003:**
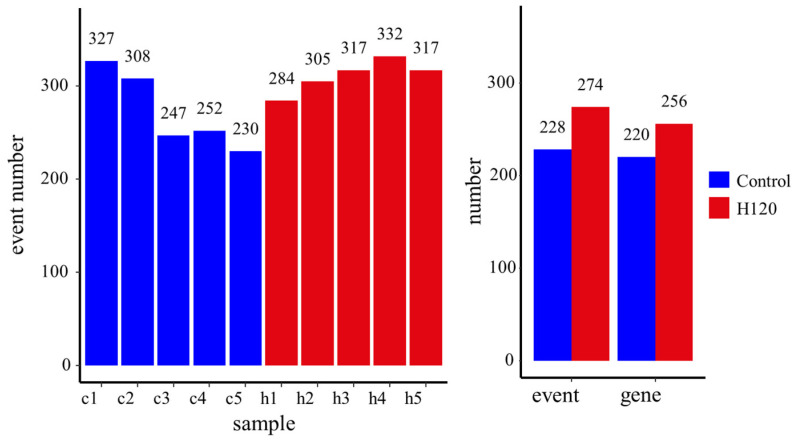
The alternative splicing events identified at the sample level and the group level. Samples c1–c5 are liver tissue samples of rats in the Control group and samples h1–h5 are from the H120 group.

**Figure 4 genes-13-00358-f004:**
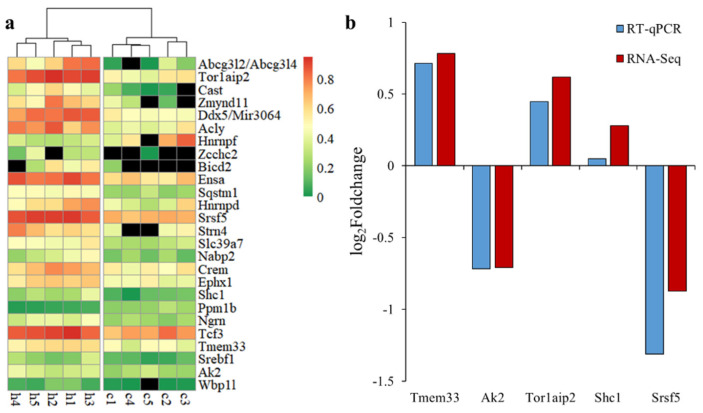
The expression mode of differential alternative splicing (DAS) and its expression verification. (**a**) pheatmap of percent spliced in (PSI) values of one variant for each DAS event in the liver. The genes (top to bottom) are sorted by differences in PSI values from large to small. (**b**) Comparative analysis of the expression level of randomly selected DAS genes in the liver using RNA-Seq and RT-qPCR.

**Figure 5 genes-13-00358-f005:**
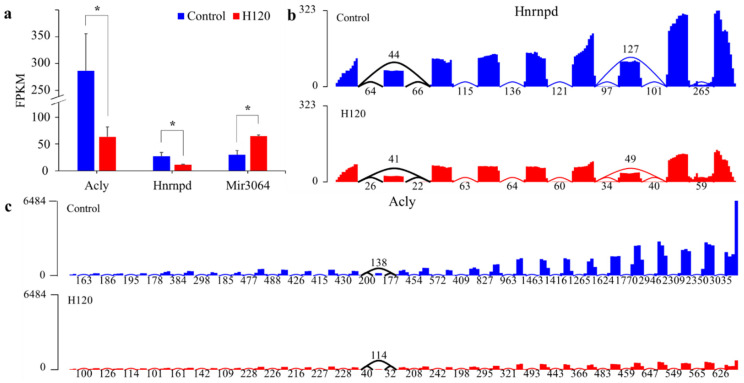
Expression levels of DAS genes and visualization of their splicing events. (**a**) The expression levels of DAS genes with different expression levels in livers in the Control and H120 groups. The asterisks (*) indicate a *q*-value < 0.05 and |log_2_FoldChange| > 1. (**b**,**c**) The average per-base read coverages (*y*-axis) and splice junction counts (labels) of *Hnrnpd* and *Acly* in livers from the Control and H120 groups. The black curve indicates the location of the differential alternative splicing event.

**Table 1 genes-13-00358-t001:** The number of all types of alternative splicing events in Control and H120 groups.

	A3SS	A5SS	MXE	RI	SE	AS	AE	Unknown
Control	28	32	8	32	47	53	26	6
H120	31	35	11	31	57	76	28	8
Increase (%)	10.71	9.38	37.50	−3.12	21.28	43.40	7.69	33.33

A3SS = alternative 3′ splice site, A5SS = alternative 5′ splice site, MXE = mutually exclusive exons, RI = retained intron, SE = skipped exon, AS = alternative start, AE = alternative end. Increase (%) = (the number of events in H120 group—the number of events in Control group)/the number of events in Control group.

**Table 2 genes-13-00358-t002:** Summary of information for the differential alternative splicing events.

Gene Symbol	Start	End	Type	Transcription	*q*-Value	dPSI
*Strn4*	chr1:78756007:+	chr1:78756355:+	A3SS:P	NM_001107480	0.0481	0.20
*Strn4*	chr1:78756007:+	chr1:78756355:+	A3SS:D	NM_001161809	0.0494	−0.20
*Ngrn*	chr1:142050672:+	chr1:142051084:+	RI:E	NM_001033900	0.0031	−0.16
*Ngrn*	chr1:142050672:+	chr1:142051084:+	RI:R	NR_028055	0.0031	0.16
*Wbp1l*	chr1:266401987:+	chr1:266409945:+	A5SS:D	NM_001127484	0.0024	−0.10
*Wbp1l*	chr1:266401987:+	chr1:266409945:+	A5SS:P	NM_001313908	0.0024	0.10
*Shc1*	chr2:188745503:+	chr2:188748894:+	AS	NM_053517	0.0048	−0.17
*Shc1*	chr2:188748359:+	chr2:188748894:+	AS	NM_001164060	0.0048	0.17
*Ensa*	chr2:197756356:+	chr2:197759882:+	AE	NM_021842	0.0000	−0.24
*Ensa*	chr2:197756356:+	chr2:197758162:+	AE	NM_001033974	0.0000	0.24
*Cast*	chr2:1506234:−	chr2:1501715:−	SE:S	NM_001033716	0.0000	0.39
*Cast*	chr2:1506234:−	chr2:1501715:−	SE:I	NM_001033715 NM_053295	0.0000	−0.39
*Hnrnpf*	chr4:149957206:+	chr4:149970689:+	AS	NM_001037287	0.0051	−0.31
*Hnrnpf*	chr4:149970237:+	chr4:149970689:+	AS	NM_001037285 NM_022397	0.6048	0.08
*Hnrnpf*	chr4:149970567:+	chr4:149970689:+	AS	NM_001037286	0.0168	0.23
*Ak2*	chr5:147200851:+	chr5:147204050:+	AE	NM_001033967	0.0165	−0.10
*Ak2*	chr5:147200851:+	chr5:147201014:+	AE	NM_030986	0.0165	0.10
*Ppm1b*	chr6:8261060:+	chr6:8271055:+	AE	NM_001270619	0.0165	0.14
*Ppm1b*	chr6:8261060:+	chr6:8273549:+	AE	NM_033096	0.8322	0.04
*Ppm1b*	chr6:8261060:+	chr6:8280124:+	AE	NM_001270620	0.0000	−0.17
*Srsf5*	chr6:104611145:+	chr6:104612019:+	A5SS:D	NM_001195506	0.0000	−0.21
*Srsf5*	chr6:104611145:+	chr6:104612019:+	A5SS:P	NM_019257 NM_001195505	0.0000	0.21
*Tcf3*	chr7:12164343:+	chr7:12167727:+	Unknown	NM_001035237	0.0013	0.15
*Tcf3*	chr7:12164343:+	chr7:12167727:+	Unknown	NM_133524	0.0013	−0.15
*Nabp2*	chr7:2825608:−	chr7:2825124:−	AS	NM_001244819	0.0261	−0.18
*Nabp2*	chr7:2825498:−	chr7:2825124:−	AS	NM_001034939	0.0261	0.18
*Sqstm1*	chr10:35713296:−	chr10:35704728:−	AE	NM_175843	0.0000	−0.24
*Sqstm1*	chr10:35713296:−	chr10:35713103:−	AE	NM_181550	0.0000	0.24
*Srebf1*	chr10:46582854:−	chr10:46579444:−	AS	NM_001276708	0.0168	−0.14
*Srebf1*	chr10:46593009:−	chr10:46579444:−	AS	NM_001276707	0.0172	0.14
*Acly*	chr10:88419161:−	chr10:88413611:−	SE:S	NM_001111095	0.0000	0.33
*Acly*	chr10:88419161:−	chr10:88413611:−	SE:I	NM_016987	0.0000	−0.33
*Ddx5*	chr10:94988461:−	chr10:94982051:−	AS	NM_001007613	0.0000	−0.34
*Mir3064*	chr10:94982117:−	chr10:94982051:−	AS	NR_128673	0.0000	0.34
*Zcchc2*	chr13:26000532:+	chr13:26014926:+	AE	NM_001122677	0.0013	−0.29
*Zcchc2*	chr13:26000532:+	chr13:26000769:+	AE	NM_001271042	0.0015	0.29
*Tor1aip2*	chr13:73708912:+	chr13:73718239:+	SE:S	NM_001165897	0.0000	0.39
*Tor1aip2*	chr13:73708912:+	chr13:73718239:+	SE:I	NM_001165896	0.0000	−0.39
*Ephx1*	chr13:99287887:−	chr13:99284094:−	AS	NM_012844	0.0000	0.18
*Ephx1*	chr13:99300580:−	chr13:99284094:−	AS	NM_001034090	0.0000	−0.18
*Hnrnpd*	chr14:11256779:+	chr14:11268562:+	SE:S	NM_001082539 NM_001082541	0.0403	0.23
*Hnrnpd*	chr14:11256779:+	chr14:11268562:+	SE:I	NM_001082540 NM_024404	0.0403	−0.23
*Abcg3l4*	chr14:5794507:−	chr14:5607839:−	MXE	NM_001037205	0.0022	0.51
*Abcg3l2*	chr14:5794507:−	chr14:5607839:−	MXE	NM_001014133	0.0022	−0.51
*Tmem33*	chr14:42540006:−	chr14:42539079:−	RI:E	NM_001034198	0.0037	−0.14
*Tmem33*	chr14:42540006:−	chr14:42539079:−	RI:R	NM_021671	0.0037	0.14
*Crem*	chr17:57082726:+	chr17:57088162:+	A3SS:P	NM_001110860 NM_001271247 NM_001271246 NM_001271102	0.0024	0.18
*Crem*	chr17:57082726:+	chr17:57088162:+	A3SS:D	NM_001271101 NM_001271245 NM_017334 NM_001271248	0.0024	−0.18
*Bicd2*	chr17:15677426:−	chr17:15675690:−	RI:E	NM_198765	0.0048	−0.26
*Bicd2*	chr17:15677426:−	chr17:15675690:−	RI:R	NM_001033674	0.0048	0.26
*Zmynd11*	chr17:63831060:−	chr17:63830296:−	A5SS:P	NM_203367 NM_203369	0.0007	0.35
*Zmynd11*	chr17:63831060:−	chr17:63830296:−	A5SS:D	NM_203366 NM_203368	0.0007	−0.35
*Slc39a7*	chr20:3822725:−	chr20:3822427:−	RI:E	NM_001164744	0.0000	0.19
*Slc39a7*	chr20:3822725:−	chr20:3822427:−	RI:R	NM_001008885	0.0000	−0.19

For the event SE, the suffixes I and S indicate include and skip, respectively. For the event RI, the suffixes E and R indicate exclusion and retention, respectively. For the event A5SS and A3SS, the suffixes P and D indicate the use of proximal (shortened intron) and distal (extended intron) splice sites, respectively. dPSI > 0 means that the average PSI in the H120 group is larger than the average PSI in the Control group.

**Table 3 genes-13-00358-t003:** Functional enrichment results of differentially alternatively spliced genes.

Group	ID	Description	*p*-Value	Count	Genes
BP	GO:0001889	Liver development	0.0000	5	*Cast*, *Ephx1*, *Ak2*, *Hnrnpd*, *Srsf5*
	GO:0032868	Response to insulin	0.0072	3	*Srebf1*, *Shc1*, *Srsf5*
	GO:0032869	Cellular response to insulin stimulus	0.0094	3	*Srebf1*, *Shc1*, *Srsf5*
	GO:0000122	Negative regulation of transcription from RNA polymerase II promoter	0.0109	5	*Srebf1*, *Ddx5*, *Tcf3*, *Zmynd11*, *Sqstm1*
	GO:0008610	Lipid biosynthetic process	0.0178	2	*Srebf1*, *Acly*
	GO:0014070	Response to organic cyclic compound	0.0415	3	*Srebf1*, *Shc1*, *Ephx1*
	GO:0050796	Regulation of insulin secretion	0.0480	2	*Srebf1*, *Ensa*
CC	GO:0005783	Endoplasmic reticulum	0.0051	6	*Cast*, *Srebf1*, *Tmem33*, *Tor1aip2*, *Slc39a7*, *Sqstm1*
	GO:0005737	Cytoplasm	0.0152	14	*Cast*, *Srebf1*, *Shc1*, *Crem*, *Ak2*, *Bicd2*, *Acly*, *Hnrnpf*, *Ensa*, *Nabp2*, *Srsf5*, *Tcf3*, *Sqstm1*, *Zcchc2*
	GO:0005634	Nucleus	0.0219	13	*Cast*, *Srebf1*, *Ddx5*, *Shc1*, *Crem*, *Ak2*, *Ngrn*, *Hnrnpf*, *Nabp2*, *Hnrnpd*, *Srsf5*, *Tcf3*, *Zmynd11*
	GO:0005654	Nucleoplasm	0.0307	7	*Acly*, *Ddx5*, *Hnrnpf*, *Nabp2*, *Hnrnpd*, *Slc39a7*, *Zmynd11*
	GO:0005789	Endoplasmic reticulum membrane	0.0419	4	*Srebf1*, *Tmem33*, *Ephx1*, *Tor1aip2*
MF	GO:0051721	Protein phosphatase 2A binding	0.0015	3	*Shc1*, *Ensa*, *Strn4*
	GO:0005515	Protein binding	0.0073	8	*Cast*, *Ddx5*, *Shc1*, *Hnrnpd*, *Srsf5*, *Tcf3*, *Sqstm1*, *Strn4*
	GO:0044822	Poly(A) RNA binding	0.0246	6	*Cast*, *Ngrn*, *Ddx5*, *Hnrnpf*, *Hnrnpd*, *Srsf5*
	GO:0003682	Chromatin binding	0.0290	4	*Srebf1*, *Hnrnpd*, *Tcf3*, *Zmynd11*

BP = biological process, CC = cellular component, MF = molecular function. The FDR means false discovery rate.

## Data Availability

The sequence data has been submitted to the NCBI/SRA database under accession number SUB6546585.

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
