# Peer review of "Analysis of Genomic Alternative Splicing Patterns in Rat under Heat Stress Based on RNA-Seq Data"

_genes, 2022, doi:10.3390/genes13020358_

Round 1

Reviewer 1 Report

This manuscript explores the molecular regulation mechanism of heat stress in mammals via conducting a comprehensive profile of alternative splicing in heat-stressed rats . this study provides reference data for the research on heat tolerance in mammalian livestock . I think that the manuscript is convenient with the scope of the journal. The paper could provide information of interest in this field, However, there are minor issues that the authors need to address before the manuscript can be considered for publication.

Lin 93- 104 : RNA extraction, cDNA library construction and Illumina deep sequencing : this section needs to rewrite and authors could use this reference as a guide (  Kang etal.2017 https://doi.org/10.3389/fgene.2017.00204 .) and this also , (Madkour etal.2021 https://doi.org/10.1080/1828051X.2021.1890645 .)

Authors should provide evidence for the condition they used (42°C for 120 minutes ) is a stress condition

 Authors should measure heat stress markers for heat stress in the blood such as corticosteroid hormone

Reviewer 2 Report

Comments

These comments are associated with the article entitled “Analysis of genomic alternative splicing patterns in rat under heat stress based on RNA-Seq data”.

It was an interesting, well written study. I would like to thank the authors for the good work done. Indeed, the response of many organs (i.e. liver) to heat stress conditions is a matter most significant due to increase of global temperature. I propose the publication of the article after “minor revision”.

I will focus on 2 different points:

Firstly, the definition of heat stress includes the combination of temperature and humidity. Many publications, on both monogastrics and ruminants, highlight the importance of humidity on heat stress conditions. The combinations of 25°C temperature and 85% humidity or 40°C and 50%, respectively are defined as heat stress. Thermal-humidity index (THI) is the most proper parameter to express heat-stress conditions. Therefore, I recommend the authors to comment of this important issue, especially in the introduction and discussion section and to add relative values if possible.

Secondly, heat stress is a situation where animals are exposed many hours per day, especially last years. In you study, rats exposed to 42°C for only 120 min. It is known that at the start of a phenomenon (i.e. heat stress) many organisms react intensively (over or down regulation of genes) but gradually the organisms come to an homeostasis and thus, the over/down regulation of many genes stabilizes. Please, make some comments (or a new paragraph) in the discussion section, emphasizing on practical interest.

Good luck!
